# The Evolution of Single-Cell RNA Sequencing Technology and Application: Progress and Perspectives

**DOI:** 10.3390/ijms24032943

**Published:** 2023-02-02

**Authors:** Shuo Wang, Si-Tong Sun, Xin-Yue Zhang, Hao-Ran Ding, Yu Yuan, Jun-Jie He, Man-Shu Wang, Bin Yang, Yu-Bo Li

**Affiliations:** Tianjin State Key Laboratory of Modern Chinese Medicine, School of Chinese Materia Medica, Tianjin University of Traditional Chinese Medicine, Tianjin 301617, China

**Keywords:** single-cell RNA sequencing, technological innovations, applications, perspectives

## Abstract

As an emerging sequencing technology, single-cell RNA sequencing (scRNA-Seq) has become a powerful tool for describing cell subpopulation classification and cell heterogeneity by achieving high-throughput and multidimensional analysis of individual cells and circumventing the shortcomings of traditional sequencing for detecting the average transcript level of cell populations. It has been applied to life science and medicine research fields such as tracking dynamic cell differentiation, revealing sensitive effector cells, and key molecular events of diseases. This review focuses on the recent technological innovations in scRNA-Seq, highlighting the latest research results with scRNA-Seq as the core technology in frontier research areas such as embryology, histology, oncology, and immunology. In addition, this review outlines the prospects for its innovative application in traditional Chinese medicine (TCM) research and discusses the key issues currently being addressed by scRNA-Seq and its great potential for exploring disease diagnostic targets and uncovering drug therapeutic targets in combination with multiomics technologies.

## 1. Introduction

The transcriptome, the whole repertoire of transcripts of a particular tissue or cell at a certain stage of development, is a collection of all mRNAs [1]. Transcriptome sequencing is the sequencing of a cDNA library of all RNA transcripts in a cell to monitor the overall transcriptional activity of a tissue or cell at the single nucleotide level, which can be utilized to not only analyze the structure and expression level of transcripts, but also to discover certain unknown and rare transcripts and precisely identify variable shear sites. Accordingly, transcriptome sequencing greatly promotes researchers’ understanding of cellular gene expression and regulation and shows great potential in disease diagnosis, clinically personalized treatment, and the elucidation of drug action mechanisms.

Cell division and differentiation trigger variability in genetic information, which is the underlying cause of the heterogeneity that exists between cells. Traditional RNA-Seq can only detect the average gene expression of the cell population; it cannot identify the heterogeneity between cells and can easily ignore the specific expression of cell subsets [2,3]. With the continuous progress of science and technology, high-throughput single-cell transcriptome sequencing methods have been gradually developed. The new methods have enhanced the throughput and detection limits of transcriptome sequencing, enabling unbiased, high-throughput transcriptome analysis of individual cells. They have become a new technology for revealing tissue composition and gene regulatory relationships.

Single-cell sequencing extends the study of genes and their functional alterations to the single-cell level. Somatic mutations, including single-nucleotide variations (SNVs) and copy-number variations (CNVs) [4], are usually rare and specific but can now be detected as such. The accumulation of DNA mutations in somatic cells is thought to be one of the causes of aging. For example, genome-wide somatic SNVs (sSNVs) in prefrontal cortical and hippocampal neuronal DNA were revealed using whole genome sequencing at the single-cell level. Neuronal sSNVs accumulate slowly with age in the normal human brain and more rapidly in early-onset neurodegeneration due to genetic disruptions in DNA repair. In addition, neuronal sSNVs are increased in both the prefrontal cortex and hippocampus, with a higher rate of increase in the hippocampus [5]. The accumulation of somatic mutations is associated with age and inter-disease molecular signaling. Genome-wide sequencing at the single-cell level identifies sSNV-rich genes in endocrine cells associated with oxidative damage with specific alternative markers [6]. In conclusion, combining single-cell transcriptional information with genomic information can identify mutational behaviors, such as single-nucleotide variants, reveal genes associated with cellular heterogeneity during disease development, enhance the throughput and accuracy of single-cell transcriptome sequencing technologies, and also provide deeper insights into clinical mechanisms.

This paper reviews the latest development of scRNA-Seq technology, including the development, and the latest research results of scRNA-Seq in the life science and medicine research fields, and discusses the possibility of its application in the TCM research field.

## 2. Evolution of scRNA-Seq Technologies

The total amount of RNA in a single mammalian cell is about 10 pg, of which only 1–5% is transcriptomic RNA, which is far from the minimum standard for single-cell transcriptome library construction [7]. To profile the gene expression activity in cells, transcriptomes require a larger total amount of starting RNA. In addition, traditional transcriptome sequencing techniques introduce amplification bias and loss of nucleic acid information in the process of mRNA reverse transcription to form cDNA. Therefore, applying the transcriptome to single-cell analysis needs to address two major issues: obtaining single cells and amplifying single-cell cDNA for library construction. The workflow of scRNA-Seq includes the following parts: single-cell isolation, reverse transcription, cDNA synthesis, single-cell library, high-throughput sequencing, and data analysis [8] (Figure 1).

Compared with traditional RNA-Seq, scRNA-Seq has great advantages in revealing the expression of some cell subsets at the molecular level, screening early diseases, evaluating clinical conditions, and clarifying the mechanism of drug action. In 2009, Tang [9] used scRNA-Seq for the first time to analyze the cDNA expression profile of a single blastomere of four-cell stage embryo mice and identify abnormal expression genes in oocytes and blastocysts, which laid a foundation for the subsequent development of scRNA-Seq. Regarding the two core aspects of scRNA-Seq, obtaining single cells and amplifying single-cell cDNA for library construction, researchers have been experimenting with technological innovations in recent years, establishing a series of sequencing technologies such as STRT-seq, Smart-seq, CEL seq, Fluidigm C1, Smart-seq2, MARS-seq, Cyto-seq, Drop-seq, inDrop-seq, 10× Genomics, Seq-Well, Smart-seq3, and VASA-seq [9,10,11,12,13,14,15,16,17,18,19,20,21,22,23,24,25,26,27]. The principles and advantages and disadvantages of the different methods are presented (Table 1).

Single-cell separation is the most critical step in scRNA-Seq. Traditional cell separation methods include serial dilution, micro-manipulation, fluorescence-activated cell sorting (FACS), and laser capture microdissection (LCM) [7,28]. With the creation of high-throughput sequencing platforms, the application of cell sorting platforms based on magnetically activated cell sorting (MACS) and microfluidics has greatly improved the efficiency, scale, and precision of single-cell isolation [29,30]. Microfluidics, a new highly integrated system with individual steps reacting on a micron-sized chip, is also called “lab-on-a-chip”. At present, this is the preferred method for single-cell separation on a transcriptome platform and is mainly divided into integrated fluidic circuits (IFCs), microporous methods, and droplet methods [14,17,25,31,32,33,34].

The construction of a cDNA library is the core of scRNA-Seq. At present, PCR-based amplification, in vitro transcription amplification (IVT), and Phi29 polymerase replication are mainly used among them, and PCR-based cDNA amplification is currently the predominant library construction method, including the end-tailing and template-switching methods [35]. The end-tailing method is speedy but causes amplification error, and the termination of reverse transcriptase reaction can reduce the coverage rate of 5′ end of transcription. The template-switching method is one of the commonly used cDNA amplification methods and can reduce the rate of nucleic acid loss, but its sensitivity is lower than the end-tailing method. IVT [31] using cDNA double strands as a T7 polymerase template is a linear amplification process, but the amplification efficiency is inefficient and IVT tends to cause amplification bias at the 3′ end of the cDNA. In order to solve the problem of amplification bias, researchers have developed unique molecular identifiers (UMIs) [36], which are used to label every single cell during reverse transcription to achieve accurate quantification of transcripts. The Phi29 polymerase replication method is a continuous amplification of cDNA using Phi29 polymerase after reverse transcription of RNA to form cDNA. This method is capable of forming complete cDNA strands during amplification, is suitable for longer sequence runs, and can enhance single-cell amplification efficiency when used in conjunction with a microfluidic system [37]. However, it was found to produce incomplete expression of low-abundance mRNAs when cells were treated and tended to result in sequence loss at the 3′ end of mRNA [38].

Analysis of scRNA-Seq data typically includes sequencing fragment alignment, cell quality control (QC), quantification, data normalization, removal of confounding factors, dimensionality reduction and feature selection, cluster analysis, and downstream analysis of the data. The raw data are first generated in the “fastq” format utilizing tools such as TopHat2, HISAT, and STAR [39,40,41,42], and QC is completed with FastQC, SinQC, and Scater [43,44,45]. Inter/intra sample normalization was carried out using sctransform, BASiCS, scran, Linnorm, etc., to offset data bias due to cell separation, mRNA amplification, etc. [46]. Then, the data were submitted to batch correction using MNNs, CCA, kBET, REBET, etc. [47,48,49,50]. Finally, dimensionality reduction was performed to resolve high-dimensional experimental data, and downstream analyses such as pseudo-temporal analysis and differential expression analysis were performed to reveal the functional and biological significance of the target cell subpopulations [51,52,53,54,55].

With a large number of scRNA-Seq methods being reported, the selection of the appropriate method to complete sequencing among the many scRNA-Seq methods has become a primary concern for researchers. The 10× Genomics method has gradually the primary choice for most scRNA-Seq due to its high throughput and low cost. In 2016, The 10× Genome company developed a novel scRNA-Seq platform based on a microfluidic droplet system, which is based on GemCode technology [56]. The platform combines cell suspensions and gel beads in emulsion (GEMs) containing reverse transcription reagents and barcoded oligonucleotides forming “GEMs” in an eight-channel microfluidic chip. Subsequently, the GMEs were lysed and their oligonucleotides were released for cDNA amplification, and cDNA libraries containing UMIs were prepared for eventual introduction into the Illumina platform for scRNA-Seq analysis. The researchers found UMI mapping rates of 38% and 33% for human and mouse (3T3) cell mixtures, respectively, which is comparable to the previously reported scRNA-Seq system. The results of principal component analysis (PCA) performed on a mixture of 293T and Jurkat cell lines at different ratios confirmed the ability of the system to perform unbiased assays on rare single cells. Subsequent clustering analysis of PBMCs identified three major cell subpopulations and several small cell clusters and generated transcriptome profiles to verify heterogeneity between 68K and PBMCs, demonstrating that this method has the capability to parse the function between immune cell populations. Finally, the genotypes of bone marrow mononuclear cells (BMMCs) after transplantation were compared with the host genes, and the differences in cell subsets between healthy people and patients with acute myeloid leukemia (AML) were revealed, which provided new insight into the analysis of cell composition and cell interaction in vivo.

To test the effectiveness of 10× Genomics, Wang et al. [16] used two transcriptome sequencing methods to analyze and compare hepatocellular carcinoma cells and found that compared with Smart-seq2, 10× Genomics had lower sensitivity and a higher dropout rate, which indicated that Smart-seq2 had certain advantages in detecting genes with low expression level, but the high throughput of 10× Genomics and the improvement of single-cell capture rate could offset the ineffectiveness caused by noise and provide stronger clustering. In addition, the cost of 10× Genomics is considerably lower than Smart-seq2, suggesting that the former may have a higher priority for the analysis of larger numbers of single-cell transcriptomes. Based on the above studies, 10× Genomics sequencing has becomes the primary choice for cell and cell subpopulation analysis.

While current scRNA-Seq methods are able to quantify and determine cell status with high precision, most methods accomplish cell capture and cDNA synthesis by poly(A) sequence hybridization of barcoded oligonucleotide primers and polyadenylated transcripts. The remaining sequences of polyadenylated RNA molecules cannot be detected, resulting in the inability to detect non-coding RNA and selective promoters (AP), etc. In addition, the absence of UMI triggers the accurate quantification of splicing events. To overcome these barriers, Fredrik et al. [10] invented a novel technology for scRNA-Seq called “VASA-seq” in 2022, the main process of which is to first disassemble single cells to obtain RNA with end-repair and introduction of poly(A), followed by molecular quantification using UFI. Finally, PCR was used to complete the amplification, and the process was similar to cel-seq (Figure 2). The advantage of the method is that VASA-seq is adjusted to two formats, plate and drop, with the advantage of the plate format being that it can meet most scRNA-Seq requirements, while the drop format has the advantage of capturing the flux cell subsets and reducing the operation time and cost.

To validate the superior performance of VASA-seq technology [10], researchers performed species-mixing experiments on mouse embryonic stem cells (MESCs) and human HEK293T cells, demonstrating better droplet spacing integrity with microfluidic technology. Subsequently, VASA-seq was compared with 10× Genomics, Smart-seq3, and Smart-seq-total, and the results showed that VASA-seq was able to maintain uniform coverage of the entire gene body. In addition, VASA-seq was superior to the other three methods in terms of sensitivity, gene detection rate, sequencing coverage, and new transcripts. By mapping scRNA-Seq at different stages after mouse embryo implantation, it was found that VASA-seq detected a slightly lower percentage of protein-coding transcriptome, but the capture rates of long non-coding RNAs (lncRNAs), short non-coding RNAs (sncRNAs), and transcription factors (TFs) were higher than other methods. Identification of cell subpopulations present in both VASA-seq and 10× Genomics showed that VASA-seq detected more differentially upregulated genes, indicating that VASA-seq was able to complete the expansion of known genes, and the subsequent identification of cell cycle status by detection of histone genes confirmed the ability of the technology to ensure cell cycle and cell type-specific histone assays. Finally, the ability of VASA-seq to measure alternative splicing (AS) in different cell types to probe cell type-specific gene function was demonstrated by detecting changes in stable AS in cells. In summary, VASA-seq has further advanced the single-cell field with its high sensitivity, full-length transcriptome coverage, and ultra-high throughput, while the technology is expected to be a new scRNA-Seq technology and an alternative to methods, such as 10× Genomics and Smart-seq3, with inexpensive and convenient operation.

## 3. Application Progress of scRNA-Seq

As an essential method to reveal cellular metabolites and expressed genes, scRNA-Seq is mainly used to analyze the interaction between cell subsets and gene regulation processes. At present, it is used in various research fields (Figure 3), such as embryonic, tissue and organ development [57,58], tumor [59,60], and immune system research [61].

### 3.1. Application of scRNA-Seq in Embryonic, Tissue and, Organ Development Research

#### 3.1.1. Embryonic Research

The mammalian embryonic development process is the essential molecular event in the growth of a living organism, and embryonic stem cells gradually differentiate into multiple-level cells and then form organs and tissues [62]. During development, all cells remain highly dynamic and heterogeneous. Interpretation at the single-cell level is required to reveal changes in specific genes during development in individual animals. Currently, scRNA-Seq can classify cells at various levels, track the developmental process and identify specific markers, and identify key genes associated with embryonic competence indicators by mapping the cell transcriptional profile at individual developmental stages [63,64]. Liang P. et al. [65] developed a machine learning platform HelPredictor using three feature selection methods and four algorithms to predict human embryonic lineage gene expression patterns and molecular events. This strategy reveals the detailed process of blastomere cell division into trophectoderm (TE), primitive endoderm (PE), and epiblast (EPI) cells. Through cluster analysis, genes were divided into eight clusters and a variety of key biomarkers were found, which improved the analysis rate of scRNA-Seq data and accurately predicted embryonic lineages.

The scRNA-Seq characterizes cell types at high resolution, sequencing and analyzing cells differentiated at variable rates and performing computational searches for cell-specific gene expression patterns to obtain significant marker genes [66]. Ilsley G.R. et al. [67] applied scRNA-Seq to the 16-cell stage of the Ciona embryo (a marine chordate) and used a novel computational approach to discover gene expression patterns. Firstly, as maternal mRNA levels varied considerably between unfertilized eggs of different individuals, and the variability and reproducibility of this technique needed to be assessed, the data were normalized. Subsequently, the gene expression pattern of individual genes was clustered and the two top clusters were generated to determine the on/off pattern for cell type per gene. An approach that does not require changes in parameter estimation or dispersion was used to rank the results, resulting in new cell-specific patterns, and cell-specific gene expression patterns were validated by in situ and single-cell qPCR. This provides an extremely valuable research model for embryonic development studies. Embryogenesis in mammals is dependent on glycolysis and oxidative phospho-uptake. Malkowska A. et al. [68] found that data were collected on the mammalian single-cell embryo atlas, and their different patterns of metabolic kinetics were determined by glycolysis and oxidative phosphorylation-related gene expression. According to modules in Computational Seurat 4.0 used to analyze the transcriptional programs of enzymes involved in glycolysis and oxidative phosphorylation, it is associated with glucose consumption during blastocyst formation in mice. The enrichment of oxidative phosphorylation genes is associated with high oxygen consumption in mouse blastocysts. Therefore, inferring metabolic regulation from scRNA-Seq data can overcome the technical problems of its metabolic profile during mammalian embryonic development, and combined computational analysis can be utilized to explore in vivo embryonic samples and identify differences in metabolic gene regulation, providing a novel genealogical model for studying embryonic development. Therefore, inferring metabolic regulation from scRNA-Seq data can overcome the technical problems of its metabolic profile during mammalian embryonic development, and combined computational analysis can be utilized to explore in vivo embryonic samples and identify differences in metabolic gene regulation, providing a novel genealogical model for studying embryonic development.

Autophagy is thought to play an essential role in the development of early mammalian embryonic cells [69]. To ensure normal development and differentiation of eukaryotes, autophagy-associated genes and mTOR mediate the degradation of cytoplasmic components, which are rapidly activated within the first 3–4 h after fertilization, independent of mTORC1 activity [70]. Zhang, K. et al. [71] studied the human transcriptome and metabolome patterns of autophagy-associated genes in early embryonic cells. The results showed that most autophagy-related genes, such as LC3B, PARP1, and PINK, maintained their genetic activity and expressed stably throughout embryonic development, but their dynamic changes were different at various developmental stages. This evidence indicates that the autophagy pathway is activated and plays an essential role in early human embryonic development, which lays a foundation for further study of the molecular mechanism of early human embryonic development.

Cell differentiation is an essential factor influencing the development of living organisms [72]. Hematopoietic stem cells (HSCs) are involved in the critical events influencing arterial vasculogenesis in living organisms [73]. Hadland et al. [74] used scRNA-Seq to analyze the transcriptional profile of cells derived from the aorta-gonadal-mesonephric (AGM) region in the transition of hematopoietic endothelial cells to HSCs, determined using the AGM-derived endothelial cell matrix model (AGM-EC). Hematopoietic precursor maturation signal and transcription factor Sox177, etc., and VE-Cadherin^+^CD61^+^EPCR^+^ cell subsets determined the dynamic transcriptional characteristics of HSC precursors, and pseudo-timing analysis revealed downregulation of Notch target gene Hey1/Hey2 and increased expression of Cdca7 during hematopoietic endothelial (HE) cell to HSC transformation, and revealed interactions between ligands and receptors during HSC development. Identifying cell cluster classes by scRNA-Seq demonstrated the molecular diversity of hematopoietic endothelial cells and obtained new matrix types, ultimately mapping a unique molecular picture of hematopoietic stem cells. Through this atlas, some subpopulations of cells directly related to hematopoietic genes were identified, linking the genes found so far with their corresponding specific cell types and making the hematopoietic genes substitute for the corresponding regulatory pathways of the corresponding cellular environment to elucidate the mechanism, which is helpful for the search and identification of targets.

The scRNA-Seq atlas, as a high-dimensional search space that covers multiple endodermal organs, such as human respiratory and gastrointestinal development, can be utilized to identify transient and differentiated cell states in different cell lineages, with an emphasis on transcriptional regulators associated with cell fate specification, benchmark cellular composition and molecular profiles in intestinal organoids in vitro and in vivo, quantifying similarity to primary counterparts, and identifying their non-targeted cell types [75,76]. Qianhui Yu et al. [77] identified NRG1, an intestinal stem cell ecotropic factor secreted by the subepithelial mesenchyme that promotes the maturation of intestinal epithelial stem cells in intestinal-like organs according to scRNA-Seq. In addition, the major intestinal regulator CDX2 was identified as a factor required in the specification of the human intestinal epithelial cell fate and the appropriate pattern of the intestine-associated mesenchyme. Therefore, through projecting cells onto a multi-organ atlas in development, examining the expression patterns of each organ marker gene, and identifying differences between organ characteristics of epithelial growth cells in different media, this study provides valuable information for cellular embryo development studies.

#### 3.1.2. Tissue and Organ Development Research

In addition to embryonic cell research, scRNA-Seq can characterize the development of distinct cell populations in organs or tissues, providing ideas for understanding some of the critical changes in the development of animal tissues and organs [78,79]. ScRNA-Seq has been reported as a developmental transcriptional map of a variety of organs (Table 2), such as the brain [80,81,82], heart [83], kidney [84], and liver [85].

In conclusion, the study of scRNA-Seq has explored the spatial and temporal programs during the development of various mammalian organs, established gene expression networks during tissue and organ development, and revealed organ developmental trajectories, which lays a solid foundation for elucidating the mechanism of human organ tissue development in vivo and potentially identifying potential factors and targets for the pathogenesis of individual organs or tissues to reveal disease mechanisms and provide a partial reference for etiology and disease treatment.

### 3.2. Application of scRNA-Seq in Tumor Research

Tumors, as the primary cause of global harm, have brought a heavy burden on human health [91]. The statistics in 2022 showed that there were approximately 23.6 million cancer cases and 10 million cancer deaths worldwide in 2019, which increased by 26.3% compared with the number of cases in 2010, and the number of deaths increased by 16% [92]. The formation of tumor cells arises from the mutation of normal cells and differentiation gene mutation in organisms. The cells with mutated genes evolve to form different types of cells or different subtypes of uniform types of cells, resulting in heterogeneity between tumor cells [93].

At present, the heterogeneity of stromal cells and immune cells in tumor infiltration in the microenvironment and the molecular mechanism between them have not been clarified in depth, which makes clinical diagnosis and treatment of most cancer patients challenging [94]. ScRNA-Seq can measure the whole transcriptome with single-cell resolution, clarify the transcriptome characteristics of each cell in the tissue, reveal the tumorigenesis mechanism and cell mutation process from the molecular level, explore the development track of different subtypes of tumor cells, construct the blueprint of the tumor cell microenvironment, discover cell groups, draw cell maps, discover new biomarkers and clarify the mechanism of drug resistance, and discover new therapeutic targets [95,96]. It can also develop a combination therapy strategy to establish multiple related cell groups by targeting tumors [97] (Table 3).

#### 3.2.1. Research on the Tumor Microenvironment (TME)

Tumors are formed by cancer cells and a variety of non-cancerous cell types, which together with the extracellular matrix form the tumor microenvironment. The involvement of these cancer-related cells, components, and immune mechanisms affects the development of cancer and is also relevant to the diagnosis and treatment of patients, as well as their prognosis [114,115,116]. Stromal cells can be divided into tumor-infiltrating T-lymphocytes (TIL-T), B cells, cancer-associated fibroblasts (CAFs), tumor-infiltrating myeloid cells (TIMs), angiogenic vascular cells (AVCs), and infiltrating immune cells (IICs) [117]. They can also be utilized to develop a combination therapy strategy to establish multiple related cell groups by targeting tumors. In the work of Aoki T. [99], the phenotypic single-cell expression profile of the Hodgkin’s lymphoma (HL) specific immune microenvironment was analyzed with single-cell resolution for the first time, and a new HL-related T cell subset with significant expression function of inhibitory receptor LAG3 was identified. LAG3 ^+^ T cell population was identified as the mediator of immunosuppression. ScRNA-Seq was applied to lung adenocarcinoma, and Bischoff P. et al. [100] identified two major microenvironment patterns called N^3^MC and CP^2^E. Microenvironment marker genes and signals identified in the single-cell map have predictive value in the whole tumor map in scRNA-Seq, providing significant clinical insights for TME biology, identifying potential biomarkers for anti-cancer, and leading to the development of new therapeutic strategies for targeted therapy. In addition, single-cell transcriptomics regulates somatic mutations and epigenetic modifications in gene expression and function and identifies minor subgroups that play critical roles in disease. Screening and detecting human disease with very small numbers of cells or individual cells in the clinical setting is an important tool for early disease screening. This will help us better understand, prevent, and ultimately cure the disease.

The phenotypes produced by intercellular interaction are not only related to the ligand-receptor, but also the interacting cell type. Lin W. et al. [101] used scRNA-Seq to analyze the transcriptome of individual cells in primary tumors or metastatic biopsy tissues isolated from patients with pancreatic ductal adenocarcinoma (PDAC). Different cell types such as tumor-cells endothelial cells, CAFs, and infiltrating immune cells were identified by unsupervised clustering analysis and a new supervised classification algorithm SuperCT. Cui Y. et al. [102] performed high-precision scRNA-Seq on pituitary neuroendocrine tumors (PitNETs) and identified three normal endocrine cell types by unsupervised clustering (PIT-1, T-PIT, and SF-1, the most specific markers for tumor classification), and the differential gene SOX9 is highly expressed in tumors expressing T-PIT and SF-1, leading to PitNETs. Zhang Y. et al. [103] used scRNA-Seq to identify twelve clusters of colon cancer corresponding to six cell types, namely cancer cells, T cells, bone marrow cells, endothelial cells, fibroblasts, and B cells. Pathway enrichment analysis showed that activation of the Wnt signaling pathway can promote granulocyte migration and lead to abnormal iron death-mediated granulocyte death and colon cancer disease. Therefore, scRNA-Seq allows for the discovery of higher-resolution cellular interactions in the microenvironment, identifying new therapeutic pathways and, as a result, effective therapeutic targets and biomarkers for patient disease stratification.

CD4+ regulatory T cells (Tregs) expressing the transcription factor FoxP3 promote tumor progression by effectively suppressing anti-tumor immunity, and scRNA-Seq identified many genes associated with Tregs [118,119,120,121]. Yang L. et al. [106] found that several immune checkpoints and their ligand transcripts were upregulated in non-small cell lung cancer (NSCLC)-infiltrating lymphocytes. Tregs and expressed specific signaling molecules, such as programmed cell death-1 (PD-1) and programmed cell death-ligand 1 (PD-L1), were also studied on the surface in EGFR-mutant NSCLC. Kieffer Y. et al. [109] found that the 0/ecm-myCAF cluster in breast cancer upregulated PD-1 and CTLA-4 protein levels in Tregs, which in turn increased CAF-S1 cluster 3/TGFβ-myCAF cell content, highlighting a positive feedback loop between the CAF-S1 cluster and Tregs. Using scRNA-Seq, Chen S. et al. [104] showed significant heterogeneity in three clusters (2, 3, and 5) consisting of CD8^+^ T effector cells, with reduced lipid and amino acid metabolism and increased levels of glycolysis observed in Tregs. In addition, high levels of KLK3 were seen in clusters 3 and 5 and Tregs, and KLK3 expression was detected in all T cell subsets, suggesting that the role of extracellular vesicles may accompany different stages of tumor-infiltrating T cells. Therefore, these findings highlight the need to assess gene expression patterns of lymphocytes at tumor sites and suggest that transcriptional data from subpopulations of tumor-infiltrating lymphocytes may help to understand the dynamics of immune regulation in the tumor microenvironment, providing novel markers for predicting antitumor immune responses.

It is well known that CAFs can promote tumor escape progression and migration [122,123,124]. Significantly increased expression of epithelial–mesenchymal transition (EMT) markers in CAFs and EMT is induced by hypoxia or activated TGF-β pathways, presumably driven by upregulation of EMT-related genes in high-grade serous tubo-ovarian cancer (HGSTOC) through changes in expression in CAFs [105]. This may reflect a metastatic role between CAFs and malignant cells, thus providing clues to the underlying mechanisms of tumor invasion and tumor metastasis rates.

B cells are heavily enriched in tumor tissue; however, the types of B cells in tumor tissue and whether there are subtypes are not clear [125,126,127]. The scRNA-Seq results showed six clusters of B cells in NSCLC, namely naive B cells with TLR10 and FCRL1, plasma-like B cells with high levels of AP2A1 and AP2M1 expression, CD20^−^CD19^−^CD79A^+^CD79B^+^ B cells with high levels of AP2A1 and AP2M1 expression, CD20^−^CD19^−^CD79A^+^CD79B^+^ B cells, CD20^+^ B cells in NSCLC, TRIM21-mediated immunoglobulins B cells, and targeting immunoglobin (IgG) B cells. The results demonstrate the heterogeneity of B cells in tumor tissue [107]. Pritchett J.C. et al. [108] used mass spectrometry flow cytometry and scRNA-Seq analysis to identify B cell features found in angioimmunoblastogenic T cell lymphoma (AITL), including reduced expression of CD73 and CXCR5 key markers. Therefore, scRNA-Seq can infer B cell populations that cannot be detected by other analytical methods.

#### 3.2.2. Research on Metabolic Heterogeneity

Metabolic heterogeneity is a key early factor in cancer metastasis, and specific metabolic pathways may exhibit stage-specific effects [128,129,130]. This difference leads to differences in tumor proliferation capacity, aggressiveness, and drug sensitivity, which ultimately affect the diagnosis, treatment, and disease progression of tumor patients. Yu T.J. et al. [110] established a metabolic classifier by scRNA-Seq to classify 10 primary breast tumors at the single-cell level, identifying different cell subpopulations and finding that malignant cells were able to promote metabolic heterogeneity. In addition, CD8^+^ T cells with dysfunction at the single-cell immune level showed metabolic activity and metabolic differences with other stromal cell subtypes. Davis R.T. et al. [111] used scRNA-Seq and metabolomic pathway analysis to find that mitochondrial oxidative phosphorylation was the main pathway upregulated in micrometastases, and glycolytic enzyme levels were higher in primary tumor cells, demonstrating oxidative phosphorylation and the functional importance of chemotaxis in cancer metastasis. Therefore, the identification of cancer cells’ metabolic subtypes, with the potential to improve patient prognosis and indicate treatment response, will contribute to understanding the metabolic heterogeneity of breast cancer and provide a new perspective for revealing the energy metabolism heterogeneity of breast cancer. Zhao Y. et al. [113] explored through single-cell transcriptome sequencing and found that the metabolic organoid HCC272 could reshape the tumor microenvironment by accelerating glucose, augmenting hypoxia-induced HIF-1 signaling, and causing upregulation of NEAT1 in CD44-positive cells, thereby inducing over-activation of Jak-STAT signaling and ultimately leading to drug resistance. Consequently, to improve drug targeting, scRNA-Seq identifies rare genes associated with drug resistance in tumor patients. Therefore, identifying cancer metabolic subtypes can improve patient prognosis and indicate therapeutic response and will help to elucidate the metabolic heterogeneity of breast cancer tumors, providing a novel perspective to reveal the heterogeneity of energy metabolism in breast cancer tumors.

Tumor-associated macrophages (TAMs), derived from myeloid precursors, are immunosuppressive and tumorigenic, and are present in high levels in gliomas [131,132,133]. Zhang et al. [134] applied scRNA-Seq to demonstrate EGFR signaling from TAM-secreted ligands to tumor cell receptors and then to downstream target genes in glioma, and identified novel multilayer network biomarkers (MNBs). There is still a gap regarding the function of TAMs in vivo and how cellular activity can be harnessed to improve anticancer therapy, and the use of scRNA-Seq to find its key influencing genes and identify its potential signaling pathways could be extremely helpful for in vivo drug action and improving treatment options.

### 3.3. Application of scRNA-Seq in Immune System Research

The immune system is an essential part of the body’s internal environment and consists of immune organs, immune cells, and immune molecules, which are responsible for the recognition and removal of antigenic foreign substances from the body [135,136]. Immune cell responses include the coordinated and balanced behavior of different cells at each stage, and their heterogeneity has not yet been revealed. Therefore, the utilization of scRNA-Seq can identify immune cell subpopulations and their interrelationships, elucidate immune cell differentiation trajectories, and reveal the pathogenesis of immune diseases and the regulatory processes of the immune system after the disease.

#### 3.3.1. Research on Immune Cell Differentiation

The heterogeneity of the immune system contributes to an effective defense against many different pathogens. The response of immune cells to antigenic substances is characterized by complex and heterogeneous heterogeneity. The skin is the outermost organ of the body, preventing the entry of pathogens and protecting against environmental damage [137,138]. Currently, a detailed characterization of the human fetal skin immune system is lacking, and it is likely that many immune cell subsets have not been identified. Xu Y. et al. [139] identified bone marrow cells, early lymphocytes, and immune cells by scRNA-Seq analysis of hematopoietic cells in fetal skin (7–17 weeks estimated gestational age). The different distribution patterns of fetal skin lymphocytes suggest that the cells undergo in situ differentiation and specialization during development. At the single-cell level, lymphocyte responses are highly heterogeneous. In some lymphocytes, many of the genes known to regulate inflammatory responses have been fully activated, while in others they are only minimally activated or inactivated, and this heterogeneity in response is associated with the stochastic nature of the gene regulatory network. The finding provides a resource for further study of the skin immune system, which can help in the diagnosis and development of effective therapies for skin diseases.

Variants associated with immune-mediated diseases are enriched with enhancers and promoters, the activity of which is upregulated upon activation of CD4^+^ T cells [140]. Soskic B. et al. [141] elucidated the heterogeneity, dynamics, and transcriptional profile of human CD4^+^ T cells, by performing scRNA-Seq on more than 1000 CD4^+^ T cells in view of regulatory T cells, a subset of CD4^+^ T cells that can control T cell activation and prevent excessive inflammation. Gene expression regulation was also mapped using scRNA-Seq spanning four time points of CD4^+^ T cell activation. The finding underscores the importance of studying the regulation of gene expression in specific environments and demonstrates the positive correlation between immunosuppression and poor prognosis by mapping the immune ecological environment on a single-cell basis. The large-scale single-cell mapping deepens our insights into immune disease-mediated genetic susceptibility and further defines the importance of the immune system for targeted therapies.

#### 3.3.2. Research on the Mechanism of Immune Disease

The immune system in the process of fighting pathogens may trigger an autoantigenic response causing immune disease and resulting in damage to organs or tissues [142,143]. With approximately 80 identified immune diseases [144], the variety and complexity of mechanisms make the rapid identification of disease triggers imperative for the treatment of immune diseases. As a new high-throughput technology, scRNA-Seq can identify the state of different types of cells in the immune system, discover the expression between different cell subsets, reveal the pathogenesis and disease changes, and provide some references for subsequent treatment. Systemic lupus erythematosus (SLE) [145], a classic human immune disease, is an essential factor in the development of diseases, such as lupus nephritis, and increased type I interferon (ISG-1) signaling is an essential hallmark of the disease [146]. To reveal the pathogenic mechanism of SLE, Deng et al. [147] analyzed the data of SLE patients and healthy controls using scRNA-Seq and found 20 cell subsets, among which monocytes, B cells dendritic cells of SLE patients, and granulocytes were significantly increased, and T cell subsets were significantly decreased. The results of differentially expressed genes of type I interferon showed that granulocytes and neutrophils were the most active in ISG activity, indicating that cell subsets were mainly related to leukocyte activation, cell secretion, and pathogen infection; transcription factors IRF9, STAT1, PLSCR1, and TCF4 were highly expressed; and the expression of genes in kidney tissue in lupus nephritis is consistent with that in SLE. The study identified new cellular markers and determined immune cell subtypes that may provide a partial reference for a cure for SLE.

Systemic sclerosis (SSc) is a disabling and usually fatal autoimmune disease [148,149]. Alyxzandria M. et al. [150] identified circulating and tissue-resident T cell subsets in healthy and SSc skin utilizing scRNA-Seq. A cluster of recirculating CXCL13^+^ T cells was identified in SSc skin that expressed a T-helper follicle-like gene expression signature and facilitated B cell responses in patients with inflamed skin. In addition, the inflammatory immune mechanisms of SSc have been studied by scRNA-Seq. Elevated expression of cytokines such as IL-32, IL-26, and IL-16 was found in the serum and lesioned skin of SSc patients, and these cytokines stimulated macrophages to secrete other inflammatory cytokines (TNFα, IL6) and chemokines (IL8, CXCL2), which in turn lead to the occurrence of inflammatory immune diseases. This discovery identifies the mechanisms driving the pathogenesis of SSc.

#### 3.3.3. Research on the Regulatory Processes of the Immune System

In addition to revealing the pathogenesis of immune diseases, scRNA-Seq can reveal the regulatory processes of the organism’s immune system caused by other factors. COVID-19 is an acute respiratory syndrome coronavirus that activates the innate immune response through the cytoplasmic DNA-sensing cGAS-STING pathway, posing a significant challenge to the security of human health [151,152]. Understanding the pathogenesis of new coronary pneumonia infections is important to prevent transmission, reduce the severity of the infection, and rapidly and effectively develop new treatment strategies. To date, there have been many studies using scRNA-Seq to fully understand the mechanisms of immune cell action in COVID-19 patients.

Huang et al. [153] used scRNA-Seq to unravel the human immune process. Eight different peripheral blood mononuclear cell (PBMC) types, such as CD4^+^ T-lymphocytes, were identified. Compared to healthy subjects, COVID-19 patients had increased DCs, CD14^+^ monocytes, and MPs/platelets and decreased CD16^+^ monocytes and NKs; subsequently, five metabolic pathways, such as IFN-I and mitogen-activated protein kinase (MAPK), were found to be upregulated using STRING analysis. However, MAPK and its transcription factor FOS were downregulated during the recovery period, suggesting that MAPK downregulation is a key event in COVID-19 healing. Meanwhile, KEGG analysis showed that genes related to iron death, such as GPX4, and others were upregulated during disease and downregulated during recovery, indicating the presence of iron and lipid peroxidation during COVID-19 pathogenesis. Finally, immunofluorescence staining results showed significant upregulation of IFI27 and BST2 in COVID-19 patients as key genes for COVID-19 infection. To reveal the difference in immune response between asymptomatic and moderate patients with COVID-19, Zhao et al. [154] found increased proportions of NK, plasma B cells, and platelets in critically ill patients compared to asymptomatic patients by scRNA-Seq of PBMC in 19 subjects, resulting in the identification of 16 cell types. T cell and NK cell data showed that the proportion of effector CD8^+^ and CD4^+^ T cells was reduced in asymptomatic patients compared to moderate patients and healthy individuals. CD56^bri^CD16^−^ NK cell abundance was significantly higher in asymptomatic patients than in moderate and severe patients and showed an increase over time in moderate patients, and XCL1, XCL2, and IFNG were highly expressed in NK cells, suggesting that CD56briCD16-regulatory NK cells play a crucial role in the fight against COVID-19 infection. Subsequently, IFN-I pathway activity at the single-cell level was explored, and four stage-dependent expression patterns were identified based on NK cells and effector T cells from moderate patients. Genes such as EGR1 and NR4A1 were found to be upregulated at later stages, and the type 1 interferon (IFN-I)-related genes ISG15, MX1, and XAF1 were more expressed in early patients and decreased over time, again suggesting that reduced IFN-I is an essential marker for recovery from COVID-19. In summary, scRNA-Seq technology has led to additional scientific insights in the fight against COVID-19 and can be used to detect important biomarkers in SARS-CoV-2 and other pathogens, reveal new subtypes of viral cells, and construct cellular profiles that are more capable of precisely identifying and tracking the origin of virus-infected cells and self-attacking T cells in blood and target tissues.

In conclusion, scRNA-Seq can identify new subpopulations of immune cells at the single-cell level and reveal patterns of immune cell differentiation and development, reveal the pathogenesis and evolution of certain complex immune diseases by identifying cellular heterogeneity, and lead to novel strategies for the diagnosis and treatment of immune diseases.

## 4. Application and Prospects of scRNA-Seq in TCM

Despite the fact that research on scRNA-Seq in TCM is still in its infancy, breakthroughs have already been made in the TCM syndrome differentiation study, the discovery of mechanisms of action and efficacy of TCM, and the elucidation of the toxic mechanisms in TCM, which also highlights the significant potential of scRNA-Seq in TCM research.

### 4.1. Research on TCM Syndrome Differentiation

In clinical diagnosis, TCM has always adhered to the basic principles of “holistic concept and syndrome differentiation treatment”, and through observation, auscultation, interrogation, and palpation to understand the state of the body’s reaction and its changes at the overall level during the disease process. However, the treatment lacks data support, and new testing technologies are needed for data integration and analysis. Using scRNA-Seq technology can solve this challenge. Colorectal cancer (CRC) is one of the most common cancers of the gastrointestinal tract [13,155]. Lu et al. [156] used Smart-seq2 to isolate 662 cells from 11 primary CRC tumors and identified and validated 14 different cell clusters and 14 differential genes such as MUC2, REG4, and COL1A2. In order to elucidate the relationship between CRC cells and TCM symptoms and microenvironmental heterogeneity, the distribution of three tumor single-cell subpopulations, excess syndrome (ES), deficiency syndrome (DS), and deficiency–excess syndrome (DES) in CRC was analyzed. It was found that DS runs through the whole process of the functional evolution of CRC tumor cells, with DES in the middle stage and mainly ES in the late stage. This technology elucidates the main pathogenesis of colon cancer from the pathway of tumor cell development and provides a scientific basis for the study of tumor cells and their microcirculatory heterogeneity.

In addition, osteoarthritis (OA) is a common chronic disease of joint dysfunction and is considered by TCM to be a “paralysis”, with deficiency of “Vital Qi” being the root cause of the disease, which can be classified into four types according to the different syndromes: wind-cold, damp paralysis, Qi-blood stagnation, damp-heat paralysis, and liver-kidney deficiency [157,158]. To investigate the differences between different symptoms of OA, Quanbo et al. [159] performed an scRNA-Seq analysis of chondrocytes from ten patients and identified seven molecularly defined chondrocyte populations. Among them, effector chondrocytes (ECs) and regulatory chondrocytes (RegCs) are early chondrocytes, while prehypertrophic chondrocytes (preHTCs), hypertrophic chondrocytes (HTCs), and fibrocartilaginous chondrocytes (FCs) are late-stage chondrocytes. The pseudospace trajectory axis reveals that preHTCs are key cells in transforming value-added chondrocytes (ProCs) into HTCs and identifies genes associated with cartilage and connective tissue development in two distinct clusters of HTCs. Identifying specific genes, such as BIRC5 and CENPU, in chondroprogenitor cells (CPCs) demonstrates that CPCs mainly maintain cartilage homeostasis and cellular multidirectional differentiation. ScRNA-Seq can determine the degree of OA lesion development based on the expression of specific genes among different cell populations, elucidating the role of different cell types in the early diagnosis and treatment of OA and enhancing the scientific and standardized nature of the dialectical treatment.

### 4.2. Research on the Interaction Mechanisms of TCM

As a treasure of the Chinese nation, TCM has been dedicated to the prevention and treatment of disease for thousands of years and has proven its efficacy through clinical practice. However, TCM contains a large number of compounds with multi-component, multi-target, and multi-pathway characteristics, and the interactions between its individual components make the specific mechanisms in clinical treatment unclear. Therefore, new research strategies need to be integrated to elucidate and study the mechanism of action of TCM at a deeper level. ScRNA-Seq, one of the most popular analytical methods in recent years, can precisely identify the cells and cell subpopulations on which the drug acts and establish the gene expression profile of the TCM component after it has been applied to the cells, the transcriptional differences of which can elucidate multi-target and multi-pathway mechanisms of action of TCM [160].

Qiu, Z.C. et al. [161] proposed a four-step integrated strategy of animal models to simulate the efficacy of TCM, serum chemistry to characterize the major components of absorption, transcriptomics to discover differentially expressed genes, and network pharmacology to predict potential targets and their mechanisms. In this study, in order to elucidate the protective mechanism of osteoporosis in rats, scRNA-Seq analysis was performed on bone marrow-derived stem cells (BMSCs) isolated from rats treated with Xian–Ling–Gu–Bao (XLGB) capsules for 3 months. Eighteen representative genes, including HDC, CXCL1, CXCL2, MIRL1, and MGP, were identified and validated by PCR for real-time quantitation. XLGB can be used as an antagonist of histamine decarboxylase (HDC) to treat post-menopausal osteoporosis. In addition, XLGB can prevent estrogen-deficient osteoporosis through ERα upregulation of CXCL1/2, elucidating the anti-osteoporosis mechanism of the systemic regulatory effect of XLGB.

In early 2020, the COVID-19 virus swept through the world, posing a serious threat to human health, and TCM treatment was found to alleviate the fever symptoms of COVID-19 patients [162,163,164,165]. Wu H. et al. [166] used scRNA-Seq and virtual screening techniques, such as network pharmacology and molecular docking, to investigate the specific mechanism by which the Xin Guan Yi (XG-1) compound prevented COVID-19 from invading the lungs of organisms. Eight cell types were identified by scRNA-Seq analysis of human lung tissues, and angiotensin-converting enzyme 2 (ACE2) receptor protein was found to be highly expressed in alveolar type II (AT2) epithelial cells. A co-expression analysis of 853 herbal targets derived from network pharmacology screening yielded 12 common genes, among which PLA2G1B, SFTPD, and SLCO4C were associated with human immunity, suggesting that XG-1 may play a preventive role against viruses through immune regulation.

Hepatic fibrosis is a histological lesion caused by liver disease with complex pathological mechanisms and no specific drugs are available to treat the disease [167,168]. In order to reveal the specific mechanism of liver fibrosis in rats treated with Ganxianfang (GXF), Liu Z et al. [160] created a rat liver fibrosis model with CCl4 and treated it with GXF, and finally constructed a liver gene map of rats by scRNA-Seq, which finally identified 2384 differential genes. These genes were significantly enriched in the ECM receptor interaction map and GXF significantly downregulated Col1a1 expression, which demonstrated that GXF could alleviate the disease in rats through collagen deposition. Meanwhile, scRNA-Seq revealed that GXF could significantly inhibit the expression of macrophage-related genes, such as CCL2, and Western blot results also confirmed that GXF could inhibit macrophage regeneration and delay liver fibrosis.

Impaired glucose tolerance (IGT) is one of the major causes of diabetes [169,170]. ZuoGuiWan is a classic formula made from a blend of eight herbs to treat IGT caused by high fat and sugar. To reveal the mechanism of action of ZuoGuiWan, Liang. et al. [171] performed scRNA-Seq on mouse embryos affected by high glucose in the 2-cycle model and identified 115 upregulated genes and 174 downregulated genes when comparing the model group with the drug-administered group. Enrichment analysis revealed that high glucose affects glucose metabolism and mitochondrial function in mice, while ZuoGuiWan can induce glucose metabolism through the upregulation of oxidative phosphorylation and respiratory chain genes via the tricarboxylic acid cycle, thereby reducing glucose load-induced mortality in embryonic cells. ScRNA-Seq can enable a network-based multi-target drug design approach to discover drug targets and target–target interactions and to analyze the mechanism of action of herbal medicines for treating diseases by comparison.

### 4.3. Research on Pharmacodynamic Substances of TCM

ScRNA-Seq is also used to study the pharmacodynamic of drugs for the treatment of certain diseases. Alzheimer’s disease (AD) is a disorder characterized by age-dependent memory loss and cognitive impairment [172]. The main pharmacological active ingredients in Panax ginseng are ginsenosides Rg1 (GRg1) and Rb1 (GRb1), which can effectively treat AD [173]. Zhang S. et al. [174] used scRNA-Seq to analyze the mRNA profiles of brain mRNA in the GRg1 and GRb1-treated 7-month senescence-accelerated mouse prone 8 (SAMP8) model. GRg1 successfully identified and validated six upregulated and twelve downregulated genes, and among the twelve downregulated genes was TRPC6, a transient receptor potential canonical (TRPC) related gene and a molecular entity associated with Ca2^+^ entry activity and involved in the AD process. GRb1 successfully identified and validated eight upregulated genes and ten downregulated genes, and among the ten downregulated genes, MAPK11 is one of the most important Aβ-degrading enzymes in preventing AD pathology. Therefore, GRg1 and GRb1 may be potential therapeutic agents to stop or prevent AD progression, which provides a theoretical basis for exploring functional drugs with active ingredients in anti-AD herbal medicine.

A host of TCMs have been clinically proven to have anti-tumor effects [175]. As a high-throughput method for screening natural anticancer compounds and their target genes in TCM, scRNA-Seq provides new insights into developing alternative anti-tumor chemotherapy for tumors. Breast cancer is one of the leading causes of death among women in developing countries [176]. Kui L. et al. [177] screened 74 herbal ingredients for anticancer activity by high-throughput screening and scRNA-Seq to discover its antitumor chemical components and potential mechanisms and analyzed the differential expression and weighted gene co-expression networks (WGCNAs) to obtain key pro-inflammatory and antitumor genes FOSL1, S100A9, CXCL12, ID2, PRS6KA3, AREG, and DUSP6, which are closely associated with the development of breast cancer. In addition, by comparing the changes in gene expression in cancer-related pathways of cellular pathway lines after treatment with herbal chemical components, three herbal medicines with potential anti-cancer effects—Ricinus communis, Gentiana, and Excoecaria cochinchinensis Lour—and three key anti-cancer signals—TNF, IL 17, and NF-kappa B—were finally identified. The scRNA-Seq technology can greatly enhance the screening of active ingredients in TCM by identifying the differentially expressed genes of medicinal substances to determine their potent components and mechanisms of action and is an important cornerstone for promoting the modernization of TCM research.

### 4.4. Research on the Toxicity of TCM

The quality uniformity and safety of Chinese medicines are still bottlenecks that currently limit the rapid development of the TCM industry [178,179,180]. Currently, there is a lack of toxicity data for many toxic herbal medicines, especially for multi-base source TCM, and there is confusion about the variety of herbs marketed. Is the toxicological profile consistent across different multi-base source TCM for the same medicinal use? At present, there are two key challenges. Firstly, the information on toxicity mechanisms is fragmented, and there is a lack of comprehensive consideration of toxicity outcomes from multiple cellular perspectives. Secondly, there are significant differences in component categories and contents between species; the basis of toxic substances is unclear, and there is a lack of “equivalent toxic component groups” that can characterize the toxicity of TCM as a whole. In view of this, how to investigate the content of toxic components and the mechanism of toxicity in multi-base source TCM is a critical problem in ensuring the safety and quality control of clinical usage. The group proposed a “hazard identification of multi-genotoxic TCM based on scRNA-Seq” by taking into account the three elements, namely toxicity effects, toxicity mechanisms, and toxic substances, and considering the migration and metabolism of components and interactions, with the help of scRNA-Seq and other multidisciplinary tools (Figure 4). Firstly, the toxicity of target organ tissues due to long-term administration was clarified by animal administration, the toxicity trends and key toxicity indicators were identified by plotting the dose–toxicity curves, and the consistency and characteristics of the toxicity effects of multi-base source TCM on target organs were clarified. Subsequently, scRNA-Seq and a quantitative real-time polymerase chain reaction (q-PCR) were used to reveal the cellular targets, key cellular subgroups, and toxicity pathways of TCM in target organs at the molecular and cellular levels. Through mass spectrometry analysis, the migration and metabolism rules of toxic components for different genes were found. Surface plasmon resonance technology and virtual screening technologies, such as network pharmacology and molecular docking, were used to screen out “toxic substances”, analyze cell types, and use Western blotting to verify the toxicity pathway. Finally, a variety of analytical methods were utilized to construct an “equivalent toxic component group” capable of characterizing the toxicity of multi-base source TCM. The method constructs a “components–targets–pathways–effects” network from three aspects (toxic effects, toxic mechanisms, and toxic substances), providing technical support for the identification of multi-origin toxic TCM hazards, which has been applied in the study of the mechanisms of toxic action of TCM, such as Uncaria.

## 5. Concluding Remarks and Future Perspectives

As a novel analytical method, scRNA-Seq has played a significant role in the life sciences and medicine. The paper reviews the latest research results of scRNA-Seq in embryonic, tissue, and organ development, as well as tumor and immune system research; presents an outlook on its application in TCM; and proposes an scRNA-Seq-based research strategy in the study of the toxicity of multi-base source TCM.

As scRNA-Seq technology is widely used in a number of basic research areas and clinical trials, there are still some critical technical issues that need to be addressed. Firstly, the single-cell capture process tends to disrupt cell integrity and activity, and there is still potential to improve cell throughput and precision; secondly, sequencing generates high-dimensional data with a large amount of noise, and how to reduce the impact of noise on high-throughput data analysis is a critical issue that needs to be addressed to improve the accuracy of scRNA-Seq. Furthermore, cell differentiation and reproduction are dynamic processes, and current scRNA-Seq makes it difficult to distinguish the specific details between cell types. In addition to technical reasons, the cost of scRNA-Seq analysis remains high compared to other analytical methods, and these are the limiting factors for the development of scRNA-Seq as a routine analytical tool. Currently, scRNA-Seq is mainly used at the gene level to reveal the mechanism of action of specific regulatory genes. In the areas of disease diagnosis, personalized clinical treatment, reproductive development, and elucidation of drug mechanisms of action, more in-depth systematic studies are still needed by combining advanced technologies at the metabolic and protein expression levels, such as spatial transcriptomic, single-cell proteomic, spatial metabolomic, and mass spectrometry flow technology, to comprehensively and accurately explain life science phenomena and drug mechanisms of action. The spatial transcriptome technology can record the original spatial location of individual cells and in combination with scRNA-Seq can localize individual cells with transcriptional signatures and generate high-resolution maps of cell subpopulations, spatially elucidating the specific roles of individual cell molecule expression and specific cell subpopulations. The single-cell proteomic technology enables quantitative and qualitative analysis of proteins between individual cells and the construction of protein molecular maps, which can be coupled with scRNA-Seq to further elucidate the status and differences between different cell types. Spatial metabolomics enables the precise identification and characterization of functional metabolites in tissues and in combination with scRNA-Seq, which enables the high-resolution localization of different types and states of cell subpopulations in tissues. The combination of mass spectrometry flow technology, which enables large-scale cell counting of specific proteins, enables not only the fine sorting of cells but also the precise analysis of cellular pathways and cell reproduction cycles within cell subpopulations. The development of scRNA-Seq and its linkage with other multiomics technologies are essential not only for elucidating the growth and differentiation of individual cells and the regulatory relationships between different genes, but also for elucidating the heterogeneity of cells in complex tissues and thus for elucidating the pathogenesis of diseases and developing novel drugs.

## Figures and Tables

**Figure 1 ijms-24-02943-f001:**
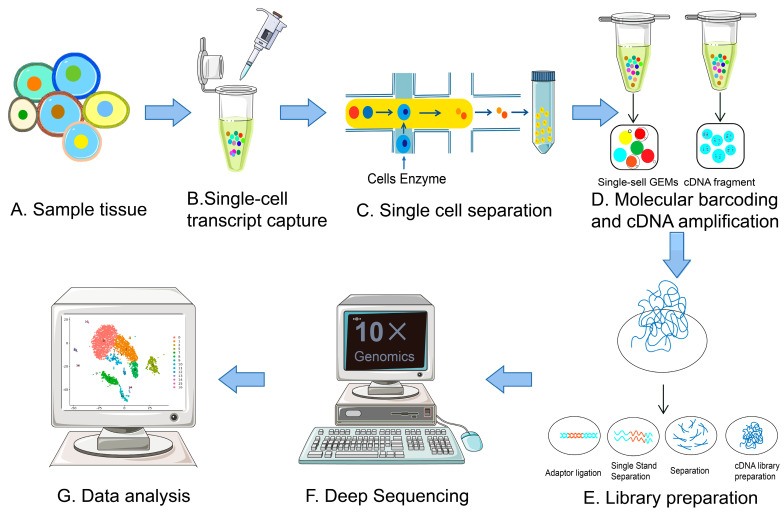
An overview of the single-cell RNA-sequencing procedures. (**A**) collect cells from tissue samples. (**B**) Single-cell capture process. (**C**) Cell isolation process. (**D**) Reverse transcription of mRNA and amplification of cDNA. (**E**) Construction of the scRNA-Seq library. (**F**) Complete deep sequencing of scRNA-Seq. (**G**) Analysis of scRNA-Seq data.

**Figure 2 ijms-24-02943-f002:**
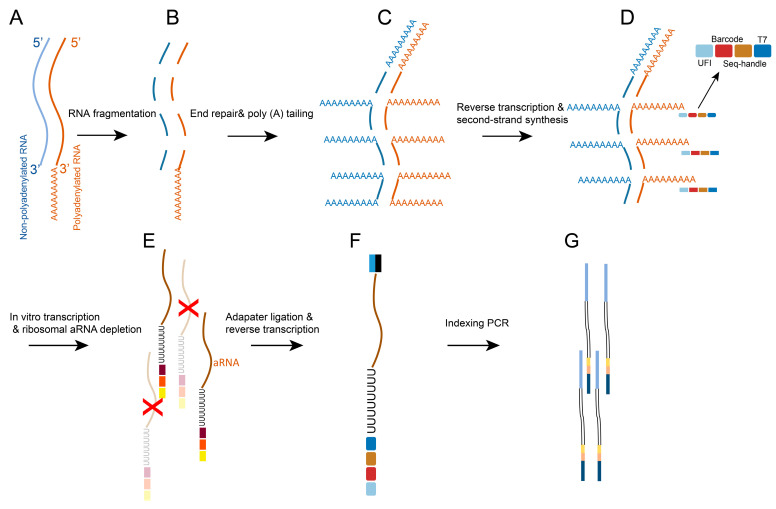
Overview of the VASA-seq workflow. (**A**) Collection of RNA in single cells. (**B**) Fragmentation process of RNA fragments. (**C**) Repair of RNA fragments and polyadenylation. (**D**) Reverse transcription of mRNA and completion of cDNA synthesis (**E**) In vitro transcription process. (**F**) Adaptation of aptamer Ligation and reverse transcription processes. (**G**) Indexing PCR.

**Figure 3 ijms-24-02943-f003:**
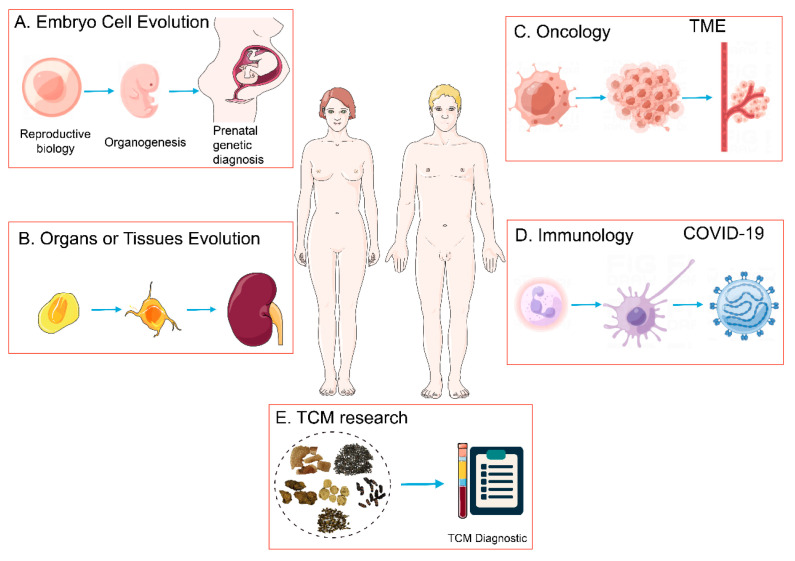
Application of single-cell RNA-sequencing technology. (**A**) Embryonic development. (**B**) Tissue and organ development. (**C**) Oncology. (**D**) Immunology. (**E**) TCM research.

**Figure 4 ijms-24-02943-f004:**
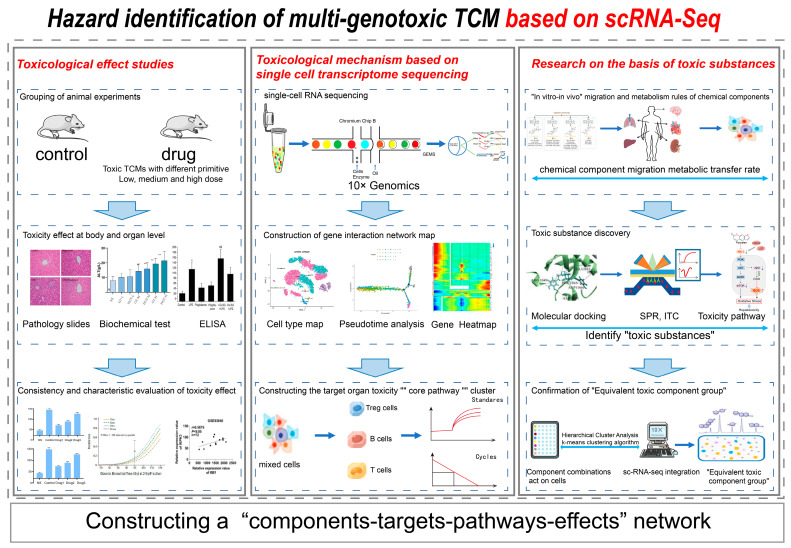
Research ideas and methods of hazard identification of multi-genotoxic TCM based on scRNA-Seq.

**Table 1 ijms-24-02943-t001:** Comparison of different ScRNA-Seq technology approaches.

Platform Name	Separation Method	Amplification Method	Using UMI	Amplification Range	Advantages	Disadvantages	Release Date	References
VASA-seq	FANS	PCR	YES	All transcripts	Low cost and accurate dosing	/	2022	[10]
Smart-seq3	Microfluidics	PCR	YES	5′ end	High sensitivity	Time-consuming	2020	[11,12]
DNBelabC4	Microfluidics	PCR	YES	All transcripts	Precise quantification	/	2019	[13]
Seq-Well	Microfluidics	PCR	YES	3′ end	Low cost and precise quantification	Unsuitable for variable splicing and allelic expression	2017	[14]
MATQ-seq	FACS	PCR	YES	All transcripts	Precise quantification	Low cell throughput	2017	[15]
10× Genomics	Microfluidics	PCR	YES	3′ end	High cell capture efficiency, fast cycle time, high cell suitability, and reproducibility	Sequencing can be performed only for the 3′ end	2016	[16]
Cyto-Seq	Microfluidics	PCR	YES	3′ end	Low cost and high throughput	Cross-contamination between RNAs	2015	[17]
SC3-seq	Micromanipulation	PCR	YES	3′ end	Good reproducibility and accurate quantification	Recognize DNA at the 3′ end	2015	[18]
inDrop-seq	Microfluidics	IVT	YES	3′ end	Low cost and linear amplification	Long operating time and high initial cell concentration	2015	[19]
Drop-seq	Microfluidics	PCR	YES	3′ end	Low cost and high throughput	Low cell capture rate	2015	[20]
MARS-seq	FACS	IVT	YES	3′ end	High specificity	Low amplification efficiency	2014	[21]
STRT-seq	Microfluidics	PCR	NO	All transcripts	Accurate positioning of transcripts at the 5′ end to reduce amplification bias	Low sensitivity, only available for identification of 5′ end DNA	2014	[22,23]
Quartz-seq	FACS	PCR	YES	3′ end	High sensitivity, reproducibility, and operational simplicity	Higher noise levels	2013	[24]
Fluidigm C1	Microfluidics	PCR	NO	All transcripts	Simple process	High cost and low throughput	2013	[25]
Smart-seq2	FACS	PCR	NO	All transcripts	Full-length cDNA detects structural and RNA shear variants	High cost, low throughput, and time-consuming	2013	[13,26]
Smart-seq	FACS	PCR	NO	All transcripts	High sensitivity to reduce the rates of nucleic acid loss	Low throughput and the existence of transcript length bias	2012	[22]
CEL-seq	FACS	IVT	YES	3′ end	Good reproducibility and highly sensitive	Low throughput and amplification efficiency, library biased toward the 3′ end of the gene	2012	[27]
Tang-2009	FACS	PCR	NO	3′ end	Good reproducibility	High cost and low throughput	2009	[9]

**Table 2 ijms-24-02943-t002:** Latest applications of ScRNA-Seq in tissue and organ research.

Tissues and Organs	Methods	Stromal Cell Subtypes	Key Difference Genes	Mechanisms	References
Liver	10× Genomics	ID3^+^ hepatocytes NCAM1^+^ cholangiocytes Sox9^+^ cholangiocytes	ID3	Inhibition of the Wnt signaling pathway maintains ID3^+^ cells in an undifferentiated hepatocyte-like state.	[85]
			COL1A1		
			HAND2		
Brain	10× Genomics	Astrocyte Radial glia cell BMP single-related cells	CLIC6	Pathways associated with metabolic stress, such as glycolysis, ER stress, and hypoxia, are significantly less activated in IVD-like organs.	[86]
			ZBTB20		
			STRN3		
Heart	STAR	Valvular interstitial cells Cardiac fibroblasts Endothelial cells	HAND1	Endocardial cells highly express ligands and receptors of the Notch signaling pathway, which regulates neuromodulin (NRG)/ERBB signaling to promote cardiomyocyte differentiation from the trabecular layer.	[87]
			HEY2		
			IRX3		
Kidney	10× Genomics	CDH1^+^/JAG1^+^ cells JAG1^+^/Jag1^+^ cells	SLC39A8	Notch pathways in JAG1 and HES1-expressing proximal/medial renal vesicles are tightly linked.	[88]
			LAMP5		
			HNF4A		
Pharynx	10× Genomics	Parathyroid mTEC cTEC	Irf628	Hippo signaling is active in the developing thymus, but absent in the Foxn1 thymus, suggesting that this pathway may function downstream of Foxn1.	[89]
			Trp63		
			Vim29		
Ovary	10× Genomics	Vascular smooth muscle cells Ovarian luminal epithelial cells Stromal progenitor cells	Foxl2l	Inhibition of BMP and Wnt signaling pathways can keep prefollicular cells in an undifferentiated stem cell-like state.	[90]
			Wnt9b		
			Nanos2		

**Table 3 ijms-24-02943-t003:** Latest applications of scRNA-Seq in cancer research.

Cancer	Method	Stromal Cell Subtypes	Key Differential Genes	Mechanisms	References
Gallbladder cancer	10× Genomics	Lymphocytes	CTLA4 TIGIT	Immunoproteins CTLA4 and TIGIT are highly expressed in CD8^+^ T cells, and bile acid and fatty acid metabolism levels are disturbed.	[98]
		Macrophages			
		Dendritic cells			
HL	10× Genomics	Macrophages	LAG3 FOXP3	Differential protein LAG3 and FOXP3^+^ T cells increase, leading to HL.	[99]
		T cells			
		B cells			
Lung adenocarcinoma	10× Genomics	Macrophages	SFTPA2	High expression of the angiogenic markers VWA1 and HSPG2 through the TGFβ and JAK/STAT signaling pathways lead to an elevated expression of genes, such as EGFR.	[100]
		NK cells	CXCL9		
		T cells	EGFR		
PDAC	10× Genomics	Endothelial cells Fibroblasts	HIF1A	The expression levels of cell type-specific markers for epithelial–mesenchymal transition (EMT^+^) cancer cells, activated fibroblasts (CAFs), and endothelial cells are strongly correlated with patient survival.	[101]
			COL1A1		
			VEGFA		
PitNETs	10× Genomics	Fibro fibroblasts	LHB	The differential gene SOX9 is highly expressed in tumors expressing T-PIT and SF-1 (P11), leading to transcriptional dysregulation in tumors.	[102]
		Endothelial cells	ZFP36		
		Immune cells	BTG1		
Colorectal cancer	10× Genomics	Fibroblasts	FABP4	The Wnt signaling pathway is activated and promotes granulocyte migration, resulting in abnormal ferroptosis.	[103]
		T cells	SPP1		
		B Cells	RBP7		
Prostate cancer	10× Genomics	T cells	KRT5 KLK3 TP63	Elevated KLK3 in T cells inhibits TNF-α, leading to prostate cancer.	[104]
		B Cells			
HGSTOC	10× Genomics	Lymphatic endothelial cells	COMP	Activation of IL6 and JAK/STAT in fibroblast and HGSTOC cancer cell subsets is involved in pathogenesis.	[105]
		Myofibroblasts	LTBP2		
		Plasma cells	TGFBI		
NSCLC	10× Genomics	CD8+ T cells	SERPINA9	Increased expression of CD54 and decreased expression of CD62L in CD8^+^ T cells led to the development of lung cancer. Furthermore, CD20^+^ B cells produced low levels of SERPINA9 and directly promoted the growth of non-small lung cancer cells.	[106,107]
		CD4+ T cells	EGFR		
		B cells	CD83		
AITL	10× Genomics	B Cells	XCL2	Upregulation of the chemokines XCL2 and XCL1 results in deranged metabolic levels of the biomarkers CD73 and CXCR5 in CD8^+^ T and AITL CD19^+^ B cell populations.	[108]
		T cells	XCL1		
		Plasma cells	CXCR5		
Breast cancer	10× Genomics	Natural killer cells	BDH2	Upregulation of aerobic glycolysis and mitochondrial oxidative phosphorylation leads to dysregulation of the metabolic level of CD8^+^ T cells and T cells.	[109,110,111,112]
		T cells	DECR1		
		B cells	PHLDA2		
Liver and biliary tumors	10× Genomics	B cells	MALAT1	The metabolically dominant organoid HCC272 can remodel the tumor microenvironment by accelerating glucose, enhancing hypoxia-induced HIF-1 signaling, and lead to upregulation of NEAT1 in CD44 cells, thereby inducing hyperactivation of Jak-STAT signaling.	[113]
		CD44 cells	NEAT1		
		HCC272 cells	SAT1		

## Data Availability

This study did not generate any unique datasets or codes.

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
