# Peer review of "The Evolution of Single-Cell RNA Sequencing Technology and Application: Progress and Perspectives"

_ijms, 2023, doi:10.3390/ijms24032943_

Round 1

Reviewer 1 Report

The manuscript entitled "The evolution of single-cell RNA sequencing technology and application: progress and perspectives" provides a comprehensive overview of single-cell RNA sequencing technology. The article is written succinctly with clean illustrations and up-to-date knowledge of single-cell RNA sequencing technology. This referee finds the manuscript in its present form to be worthy of publication and sees no major concerns. However, I feel that a section on the role of single nucleotide variations in identifying subpopulations and genotype-phenotype associations should have been provided. Although the illustrations and captions appear to be perfect, high-resolution images should be provided.

Author Response

Thanks to the reviewers for their valuable opinions. We have carefully revised the manuscript according to the opinions. Please see the attachment for detailed amendments.

Reviewer 2 Report

In this review paper, Wang and co-authors attempt a review of the scRNA-seq technologies and applications, in the context of their recent and accelerating progress. The subject is interesting and could benefit some readers, especially by covering applications in traditional Chinese medicine. However, the review is not written clearly enough and misses some conceptual developments and applications. Furthermore, the writing and the choice of words need thorough revision, including for factual inaccuracies  across the text (see examples under); while I highlight just two examples in the beginning of the paper, almost every sentence in the text can benefit of revision. In addition, the review does not go in depth, challenges assessment, and discussion but seems to mostly list some of the facts/events which does not contribute substantially to the Reader’s comprehension of the subject. Overall, I do not find the manuscript acceptable for publication.

Examples of wording issues:

1)    Intro: the 2nd sentence (“RNA sequencing (RNA-Seq) is to…”) is long, unclear and with mixed info content; also “state” is repeated, revise

2)    “All cells in a single organism have the same genetic material…” this is incorrect due to somatic variants.

3)    The sentence “Generally, it is used to identify differential gene expression and epigenetic factors caused by single cell gene mutations, and it has quickly become one of the most powerful tools for analyzing cell gene expression in the 21st century” is neither clear nor accurate.

4)    …most of the sentences and paras can benefit revision.

Author Response

(The authors gave the same response as above.)

Reviewer 3 Report

This review was well-written and provided updated knowledge on emerging sequencing technology, single-cell RNA sequencing (scRNA-Seq). The manuscript focuses on the recent technological innovations in scRNA-Seq, heightening the latest research results with scRNA-Seq as the core technology in frontier research areas such as embryology, histology, oncology, and immunology. Moreover, the authors presented the  prospects for scRNA-Seq innovative application in traditional Chinese medicine (TCM) research and discusses the key issues currently being addressed by scRNA-Seq and its great potential for exploring pathogenesis of diseases, disease diagnostic targets and uncovering therapeutic drug targets in combination with multi-omics technologies. I recommend it for publication in its present form.

Author Response

(The authors gave the same response as above.)
